# Envisioning the Future of Heritage Tourism in the Creative Industries in Dubai: An Exploratory Study of Post COVID-19 Strategies for Sustainable Recovery

**Farooq Haq** [1] , **Naveed Yasin** [2,*] and **Gayatri Nair** [2]

[1] Faculty of Management, Canadian University Dubai, Dubai 999041, United Arab Emirates; farooq@cud.ac.ae
[2] Department of Creative Industries (FCAS), Canadian University Dubai, Dubai 999041, United Arab Emirates; gayatri.nair@cud.ac.ae
[*] Correspondence: naveed.yasin@cud.ac.ae

**Abstract:** This exploratory study investigates the challenges for the heritage tourism industries in the UAE by focusing on the strategies and policies adopted during the COVID-19 pandemic. This study aims to identify the business strategies, analyze the contextual challenges for service providers, and explore how providers managed these challenges during the COVID-19 pandemic. A multi-case study approach is adopted with 12 renowned and leading heritage tourism providers (CEOc, managers, and relationship managers) Additionally, four tourism academics interviewed through qualitative semi-structured interviews. The data were obtained during the post-COVID-19 economic recovery period from January 2022 to November 2022. The protocols of the Delphi technique and the Template Analysis (TA) approach were combined to develop richer insights. Initially, the challenges discovered were thematically categorized under two levels, which were (i) Inconsistencies for Heritage Tourism and (ii) Appreciation for Heritage Tourism. As a result, subsequently, four multi-tiered themes (macro, meso, and micro level) emerged as a response to the current challenges: (i) Repackaging heritage tourism, (ii) Long-term safety measures for tourists, (iii) Organizational adaptation and innovation, and (iv) Creative recommendations. The novelty of this study is inherent in its contextualization of an under-explored area, concurrent analysis of a dynamic and lucrative sector, and methodological advancement through the embellishment of techniques. Based on the findings of this research, a contextualized framework is proposed that complements tourism theory and delivers credible implications for researchers, government planners, and tourism providers.

**Keywords:** heritage tourism; COVID-19; tourism management; creative industries; sustainable tourism

## 1. Introduction

The declaration of COVID-19 as a global pandemic by the World Health Organization officially on 11 March 2020 caused immense havoc and a global emergency, alarming the masses to protect themselves against the infectious virus [1–4]. In an emergency response to the severe health crisis, the UAE Government temporarily suspended all visas for passengers arriving from any country as of 19 March 2020 [5,6]. However, after a few months, the government re-opened the country to tourists from selected countries, where all passengers were required to take a PCR test before airport immigration upon arrival [7]. To attract tourists, this requirement was abolished in the summer of 2022. These incremental steps were decided by the government at the macro level, implemented by the Ministry of Tourism and travel groups at the meso level, and applied by travel agencies and heritage guides at the micro level [8]. The unpredictability of the COVID-19 virus has compelled organizations and governments to reorganize their operations and management, and several markets and industries had to undergo restructuring ([4,9,10]). This restructuring also recognized heritage tourism as part of the creative industries, although it was believed to be an exclusive subset of the cultural tourism industry [11,12]. The tourism industry

was most severely damaged, which was responsible for feeding several other domestic industries. The COVID-19 pandemic hit any kind or type of tourism in and out of the country, and with travel restrictions and borders closing to minimize the spread of the virus, the damage continues. The overall situation was so bleak that no organization or government could foresee or forecast the financial and business revival and the timeframe scheduled for re-opening tourism-related businesses [3,10].

The 21st century experienced a deadly outbreak of 'severe acute respiratory syndrome' (SARS), which spread in Southeast Asia, negatively impacting the travel and tourism industry, with the outbreak eventually being controlled [13]. The consequence of SARS across Asian countries was infection among tourists, demonstrating how unchecked health controls and open international travel could intensify the spread of COVID-19 [13,14]. The SARS pandemic ended up infecting less than 10,000 individuals in total. The total damage inflicted on the related countries' economic front is roughly around a GDP loss of USD 20 billion [15]. Researchers from various backgrounds analyzed the spread, damage, and control of the SARS virus, concentrating on Asian tourism. All studies concluded with the importance of health control and social prevention methods in achieving a healthier global future [13–15]. Comparatively, COVID-19 reached over six million fatalities around the world. Although infection cases are still being reported, patients' chances of dying are almost negligible from a financial perspective; the economic figures reported by the International Monetary Fund [16] and United Nations World Tourism Organization [10,17] demonstrate the sheer economic impact of COVID-19 on heritage tourism. The struggle and success of the tourism industry with contagions such as SARS have applied to COVID-19, where public and private tourism providers had to craft emergent strategies for business survival. The current research highlights some tourism opportunities that emerged from various health crises, but the threats have been more impactful [10]. Statistical data have been published indicating the damaging influence of COVID-19 on Dubai tourism and some recovery strategies from the Dubai Government. Nevertheless, no in-depth participation from tourism experts, planners, and marketers are available in the current literature on heritage tourism in Dubai [18].

In a previous study, the impact of COVID-19 on people's mobility in tourist cities related to retail, parks, and transit stations was investigated, looking at several international visitors published by MasterCard [19]. Similarly, [16] predicted that global economic growth would drop by 3 percent in 2020 due to COVID-19. Specifically, the developed world would experience an economic loss of 6.1 percent, while emerging markets and developing countries would suffer a loss of 1 percent. As a primary control mechanism for the spread of COVID-19, most countries imposed travel restrictions, resulting in 93% of destinations in Europe being closed to tourism. In the USA, the number was 82%; in Asia and the Pacific, 77%; in the Middle East, 70%; and in Africa, 60% [17,20,21]. The suspension of international flights triggered a massive crash in demand for the airline industry. The global airline business revenue was estimated to have experienced a drop of US USD 314 billion in 2020, which equates to 55% less than 2019 [17]. COVID-19 resulted in a 22% drop in the overall movement of global travelers in the first three months of 2020, which decreased by 60–80% throughout 2020 [1,22]. The travel and tourism sectors are vital for the UAE economy. In 2016 this sector provided AED 159.1 billion (USD 43.3 billion) to the UAE's GDP. This contribution represented 12.1% of the nation's GDP [18].

Heritage tourism could be understood as a type of tourism that is based on, firstly, natural heritage, including sea, rivers, mountains, beaches, and deserts, described as a subset of eco-tourism; and secondly, cultural heritage, containing the history of the country's people, including religious places, archaeological sites, and tombstones [23,24]. The tourism providers recognize heritage sites authenticated and certified by the World Heritage Committee, a subsidiary of UNESCO [25–27]. The concept and tradition of heritage tourism are as old as tourism itself, or when it was first recognized as a human practice. Heritage tourism is acknowledged as a type of tourism that is related to the heritage, religion, culture, and social traditions of a given geographical place [28–30]. The

research authors concluded that heritage is "capable of being interpreted differently within any one culture at any one time, as well as between cultures and through time" [31]. The statement from the International Council on Monuments and Sites [32] helps us to understand heritage as an all-inclusive subject that includes tangible assets such as natural and cultural environments, landscapes, historical sites, and building environments, in addition to intangible assets such as collections, past and continuing cultural norms, knowledge, and living practices [32]. Dubai has been recognized as a leading luxury tourism destination offering privileged services, authentic glamorous shopping malls, luxury hotels and resorts, warm beaches, original deserts, international cuisines, global shopping, and heritage destinations [32–34]. It has been noted that Heritage Tourism did not receive critical attention in Dubai due to the concentration on luxury-based self-indulgent tourism [33,35]. However, the value of heritage tourism in Dubai has grown massively, with many tourists showing their interest in the history and culture of this marvelous city that sparkles as a modern gem in today's world of tourism. Moreover, tourism planners have recognized that the cultural identity and authentic heritage of Dubai can establish a tourism clientele that could attract huge numbers of international heritage-based artistic enthusiasts [35–37].

To strengthen heritage tourism in Dubai, it is more suitable to focus on tangible heritage where archaeological places in Dubai are endorsed as authentic by different overseeing groups [25,36,38]. Based on the post-occupancy evaluation (POE) used to identify characteristics of the tourism-oriented heritage districts from locals and tourists, they discovered that Dubai's Shindagha Area and Al Fahidi Historical Area were considered cultural and heritage places that attracted tourists [35,38,39]. Authors argued in their research that UNESCO-certified heritage sites attracted more global tourists "because the inscription guarantees the value of heritage" [36]. Therefore, Dubai's public and private tourism providers have been respecting the requirements set by UNESCO to authenticate their natural and cultural heritage tourism sites. This process was intensively damaged by the COVID-19 pandemic, which diminished all tangible and intangible activities that had previously stimulated heritage tourism in Dubai.

Based on the importance of the heritage tourism industry's survival, this exploratory study focuses on the post-COVID-19 strategies adopted for sustainable recovery in the United Arab Emirates. This study presents the following research objectives:

1. Identify business strategies of heritage tourism at the macro, meso, and micro levels during the COVID-19 pandemic.
2. Analyze the contextual changes that were implemented for heritage tourism management strategies during and after the COVID-19 pandemic.
3. Examine how tourism organizations managed the consequences of new initiatives in response to the pandemic.
4. Propose tourism policy measures for policymakers for post-COVID-19 recovery.

## 2. Materials and Methods

Qualitative methods of inquiry were adopted for this study using the Delphi technique and the Template Analysis (TA) approach to gain valuable insights from tourism experts based on their 'lived experiences' during an unprecedented pandemic (i.e., COVID-19) and to assess the influence and impact of this phenomena. This qualitative study explores the contextual challenges associated with the pandemic, and a multi-case study approach was adopted with 12 leading heritage tourism providers and four academics as subject scholars. The data were collected from January 2022 to November 2022.

The search for specialized tourism experts and academics also highlighted the adoption of non-probability sampling, justifying a method appropriate for exploratory research, mainly used if the subject was untapped and limited [40–42]. The Delphi Technique enabled us to access an unbiased selection of tourism experts and academics based on their experiences, information, observations, and business performance in the tourism industry [32]. As indicated in Table 1, these participants were interviewed through several

qualitative semi-structured interviews. The data were obtained from CEOs, managers, and relationship managers within each (case) during the post-COVID-19 economic recovery period. The qualitative data were obtained through a snowball sampling approach initiated from networking at tourism events across the UAE.

**Table 1.** Overview of cases.

| S. No | Participants | Tourism Service Provider | | Challenges |
|---|---|---|---|---|
| Case Study A 1 | CEO and Relationship Manager | Travel and Tourism Booking Agency | 1.<br>2. | The tourism industry has become static<br>Many guides returned to their home country |
| Case Study A 2 | CEO and Manager | Culinary Travel | 1.<br>2.<br>3. | Tours and airlines were suspended<br>A large number of cancellations and refunds<br>Rework packages and offers to customers to attract them |
| Case Study A 3 | Manager and Relationship Manager | Online Travel Agency | 1.<br>2. | Health and safety concerns of customers<br>Reluctant to travel and explore due to the pandemic |
| Case Study A 4 | Travel Agent and UAE Cultural Consultant | Cultural Tourism Agency | 1.<br>2. | Shift to smaller group tours and outdoor visits<br>Drop in demand to travel and explore |
| Case Study A 5 | Travel Agent and Manager | Travel and Tourism Booking Agency | 1.<br>2.<br>3. | Longer time for visa approval<br>PCR Tests were mandatory<br>Quarantine was made mandatory for positive results that disrupted the itinerary |
| Case Study A 6 | Relationship Manager, Heritage Guide, and Local Tour Guide | Airline Tour Packages | 1.<br><br>2. | Packages needed to be more competitive and cost-efficient<br>Changes in customer preferences, such as preference for private tours |
| Case Study A 7 | Travel Agent and Local Tour Guide | Online Travel Agency | 1. | Suspension of services (hotels, flights, attractions) during the pandemic |
| Case Study A 8 | Relationship Manager, Travel Agent, and Heritage Guide | Airline Tour Packages | 1.<br>2. | Returning to normal state<br>PCR Testing and regulations |
| Case Study A 9 | Academic | Tourism Subject Expert | 1.<br><br><br>2.<br><br>3. | Lack of information regarding the medical threat of COVID-19 and future travel regulations<br>The impact of the pandemic on tourism was not measurable<br>"The demand for a degree in tourism also dropped; hence my university is closing the tourism program". |
| Case Study A 10 | Manager, Heritage Guide, and Cultural Consultant | Cultural Tourism Agency | 1.<br>2. | Financial constraints of tourists and saving pattern<br>Lack of interest in touring heritage sites |
| Case Study A 11 | Manager, PRO, and Local Tour Guide | Travel and Tourism Booking Agency | 1.<br>2.<br>3. | Change in customer preferences<br>Health and safety concerns<br>Visa approval and processing were time-consuming |

**Table 1.** *Cont.*

| S. No | Participants | Tourism Service Provider | Challenges |
|---|---|---|---|
| Case Study A 12 | Academic in Marketing | Marketing Subject Expert | 1. All market research plans were disrupted, and no data on tourists' behavior and perceptions could be collected<br>2. Most marketing agencies working on tourism products and services went out of business<br>3. The survival of heritage tourism is the best case study for market survival strategies |
| Case Study A 13 | Relationship Manager and Travel Agent | Online Travel Agency | 1. PCR testing and quarantine<br>2. Lack of heritage tour guides |
| Case Study A 14 | Manager and Local Tour Guide | Culinary Travel | 1. Financial issues and budget control problems<br>2. We already had many health regulations with food handling; now, many more will be introduced<br>3. Need to shift business focus as food-related heritage business will take years to make a comeback |
| Case Study A 15 | Academic in Economics | Economics Subject Expert | 1. Financial issues and global economic crisis<br>2. Global inflation due to hike in transportation costs<br>3. Oversupply of tourism products and services with diminishing demand |
| Case Study A 16 | Academic in Cultural Studies | Cultural Studies Subject Expert | 1. With the health and economic crisis, no one is interested in culture and heritage<br>2. Health precautions during pandemics based on cultural or herbal medication and cures could be further explored |

Source: Authors.

In the Template Analysis approach, the following themes were used to formulate the interview guide pertaining to business strategies, changes, and tourism management, managing changes at various levels, as well as policy measures and recommendations. The thematic data analysis process (TA) is presented in Figure 1. The Template Analysis (TA) involves generating codes to identify and classify the data into themes and patterns [43]. In this study, the TA approach was implemented (See Figure 1), which created themes around the research question, literature, challenges, and the strategies of the heritage tourism provider in response to the effects of the pandemic. The identified themes highlighted the key issues that provided a rich understanding of the situation and developed tools and strategies to address the challenges.

The data were collected through semi-structured interviews at offices, public spaces, and the university campus. All responses were collected as interview transcripts and analyzed individually; the findings were reconciled. To further enhance the quality of the data analysis process, direct statements provided by various tourism experts working for public and private agencies and academics were searched through government tourism websites, UAE newspapers, blogs of tourism service providers and influencers, and other social media platforms.

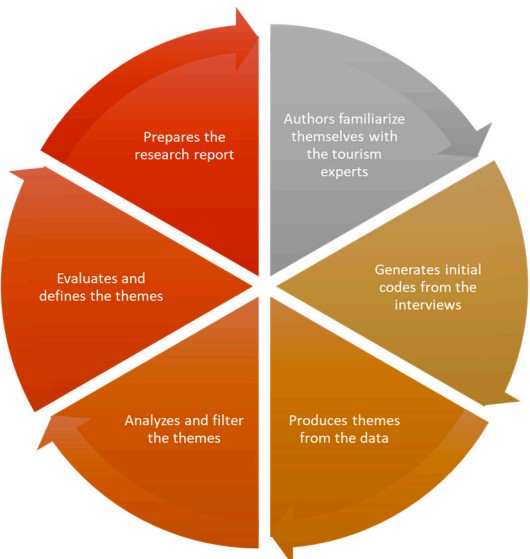

**Figure 1.** Data Analysis Approach (Template Analysis). Source: Authors.

Various new ideas and observations were noted from interviews about the heritage tourism management of COVID-19 in Dubai. Each author read individually written transcripts by each researcher numerous times to conduct the cross-case thematic investigation [41,42]. The template analysis approach based on data coding was directed by the research questions [44,45]. Adapting from [45], the primary coding for the collected data was based on evidence extracted from discussions with all respondents. Likewise, codes were written on the quoted and any other relevant information collected from online and social media resources. Successively, handwritten memos were compared for each copy allocated to each interviewee and all online resources to reach rich and credible ideas emerging from communications with tourism providers and academics. The Delphi technique uses a structured communication approach encompassing diverse opinions from a panel of stakeholders and experts. In this study, the Delphi technique gathered inputs from 12 heritage tourism providers who were asked open-ended questions and shared their opinions anonymously.

The reliability and validity of data analysis were maintained by applying the cross-case investigation to the data collected from experts, arranged individually and mapped with statements collected from online platforms [45]. The investigator triangulation was conducted when each researcher separately analyzed the information gathered from various primary and secondary sources, triangulating the emerging themes to be classified as findings of this research [30,43].

## 3. Results

The findings of this study, based on interviews with tourism experts, indicated a positive mood and confidence surrounding tourism recovery in Dubai. The impact of the pandemic on tourism and related businesses in Dubai was evident from discussions with all respondents. The tourism industry experts and academics shared several suggestions about strategies that could be adapted by companies within the heritage and tourism industry to manage their businesses during instability and unforeseen crises. The experts explained various emergent strategies crafted and implemented in tourism services and marketing that supported the recovery of heritage tourism.

Many touristic companies were shut down or moved into new business territories, while some were temporarily closed. Meanwhile, others proposed new experience-related offerings adapting to the COVID-19 restrictions implemented in the global tourism industry. As depicted in Figure 2, four major themes were identified from the Template Analysis:

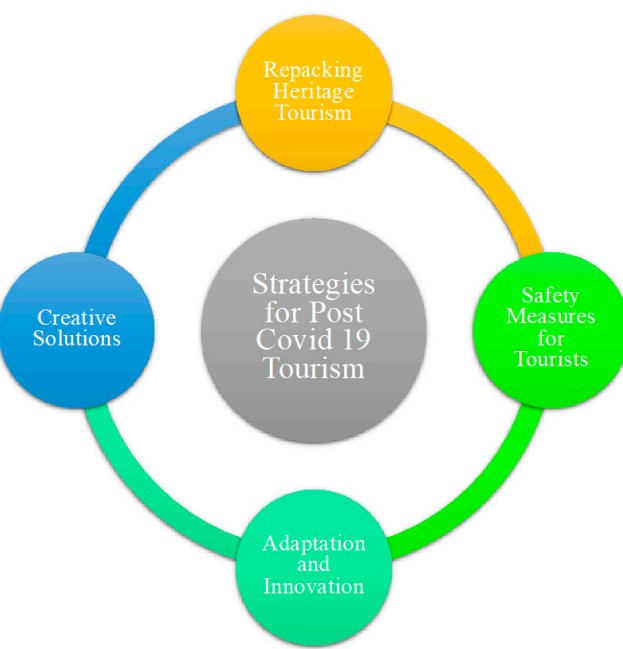

**Figure 2.** Strategies for Post-COVID-19 Tourism. Source: Authors.

### 3.1. Repackaging Heritage Tourism in Dubai

An overarching theme appreciated in this research that was recognized by all business and academic participants was repackaging heritage tourism in Dubai. Due to the rapid shutdowns caused by the travel regulations, tourism companies intensively suffered as sales dropped, and many employees were laid off. The surviving companies found an opportunity to re-pack or re-bundle their travel packages, with the variety of services offered being diversified. The diversified services were offered using internal resources or by entering partnerships with other service agencies.

The Dubai Chamber of Commerce specialist elaborated on this process: "70% of businesses would have shut down because of COVID-19, but they survived as Dubai's economy is driven by entrepreneurial leadership, service innovation, and a diversified economy. So, we found bundle packaging of Dubai tourism as the way forwards, which included heritage ad culture as key selling elements" [45].

On similar grounds, a private operator said:

"All activities have been focused on managing the pandemic. The necessary preservation and promotion of heritage experience have been seriously damaged. Additionally, social distancing measures have prevented access to heritage places hence dropping the number of visitors. Hence, my company has combined food tourism with heritage and culture to attract more travelers to Dubai".

During the pandemic, the CEO of Dubai Airports was looking for hope when he stated: "there could be quite a low level of activity for some time" [46]. As the Director of the Middle East and African Region at STR Global explained in a discussion: "More than a third of the city's 120,000 hotel rooms will remain closed in the summer months as most owners try to save on operating costs. The industry employs about 40,000 people . . . otherwise, you are asking the owners to reach into their own pockets and, while some might do that, others will not be able to afford it" [47]. Similarly, an executive in Emirates highlighted COVID-19-related redundancies in Emirates Airlines: "Given the significant impact that the pandemic has had on our business, we simply cannot sustain excess resources and have to right-size our workforce in line with our reduced operations".

Another respondent operating heritage walking tours in Dubai business explained her company's experience and solutions:

"From March 2020, tourism slowed down and froze down within weeks. All our tour guides were jobless, while many returned to their home countries due to closed borders; some even got stuck here without any income. Then, we started walking tours for local Emiratis and residents where the experience combined food, culture, heritage, shopping, and personal services. Our customers were delighted to try different cuisines and see new shops for the first time. We partnered with various specialized shops and service centers, including pet shops, hairdressers, and spas".

Similarly, the founder of an experiential company stated that:

"We were compelled to suspend our tours due to the global spread of the pandemic, especially with the enforcement of lockdown and quarantine. The tourism industry, including us, was also impacted by a huge wave of cancellations and refunds on bookings. We had to divest our services and add more services as experiential traveling for locals or few travelers".

Elsewhere, as the owner of an elite and luxury experiential travel agency expressed: "Disaster hit us in the shape of COVID-19 since we offer luxury heritage experience, and it was the last thing in anyone's mind. As an urgent strategy, we had to repackage our services with a different price range and offer more experiential services to our customers".

When Japan decided to postpone the Olympic Games, the UAE Government opted for the postponement of the EXPO 2020 till 1 October 2021, which was an event aggressively marketed with high hopes for economic growth within the country. Local and international enthusiasts had been waiting impatiently for the Expo since 2013 [17]. EXPO 2020 was supposed to bring 25 million visitors, mostly foreigners, with an estimated AED 122.6b gross value added to the UAE's economy [48]. As the Dubai Chamber of Commerce expert reported: "The government had to pass a resolution to delay the Expo 2020 Dubai, which will lead to shortfalls in expected revenue and require a revision of the economic forecast".

### 3.2. Long-Term Safety Measures for Tourists

Since the diversification of the economy of Dubai depends heavily on tourism, the authorities decided to re-analyze and re-plan the restoration of Dubai tourism by focusing on safety precautions and control measures on a long-term basis. Hence, in 2021, the Dubai Government reviewed and boldly decided to apply various precautionary strategies to re-open all types of tourism for overseas travelers.

The Dubai Government implemented safety measures that aligned with the WHO protocols focusing on personal hygiene and social distancing [49]. Moreover, tourists and staff were required to always wear masks and carry out regular body temperature checks before and after entering any site. Social distancing was compulsory, with two meters of distance between people. Any tourists with chronic or respiratory diseases were strictly prohibited from joining the tourism experience. All sites had to allocate an isolation area to quarantine any staff member or tourist showing symptoms of COVID-19.

The 'Smart Tourism' strategy adopted new technology for innovative payment and ticketing solutions for all tourists [50]. Information booklets were no longer allowed to be distributed to customers, and online information was preferred [51]. Tourism and travel companies had to frequently clean and disinfect the venue's entrance/exit gates, toilets, lifts, railings, stairs, escalators, and elevators. The Government of Dubai took the responsibility of daily intensive sterilization of all public areas, with hand sanitizers placed at the doors of all toilets and public areas, and each tool or equipment being disinfected after each use [51].

To restore tourism in early 2021, the UAE Government implemented well-measured and planned safety measures. In compliance with this strategic move, the CEO of Dubai Airports declared that: "Dubai is fully committed to following all guidelines and measures to ensure your family's health remains our top priority. From the moment you arrive at the airport to when you check in to your hotel, whether you take your family shopping, treat

them to the thrills of a waterpark, or have a relaxing day at the beach, we have ensured your well-being is safeguarded every step of the way" [45]. The UAE's efforts have paid off, as it was recognized as a safe destination on 7 July 2020 by The World Travel and Tourism Council (WTTC) [49].

An outstanding policy developed by the Dubai government was the decision to launch a compliance program to certify, recognize, and issue a "Dubai Assured Stamp". This strategy was implemented with the collaboration of the Dubai Tourism and Dubai Municipality departments. The "Dubai Assured Stamp" was awarded to hotels, tourism operators, retail establishments, food and beverages outlets, and tourist attractions that have applied public health safety steps for control and prevention of COVID-19; it is valid for 15 days and renewed every fortnight after inspection, assuring the safety of guests and employees [49].

An Emirati cultural tourism operator exemplifies that:

"The Dubai Assured stamp will be the winning punch in global tourism, demonstrating to visitors that each hotel, restaurant, retail outlet, and attraction complies with strict safety and hygiene measures, which follow the international health and safety standards and protocols".

The respondent arranging heritage walking tours also elaborated on this strategy:

"The 'Dubai Assured' stamp will be the Messiah for Dubai Tourism, awarded to all compliant businesses following necessary precautionary measures. Several government inspectors will check the implementation of these measures to renew the stamp every 15 days. This stamp is also being utilized as a marketing tool informing the global tourist that Dubai visitors are safe, and we care for them".

Likewise, the local experiential company owner said that:

"Now we have mini groups of a maximum of eight people for the city heritage tour. It works like a bubble where we can keep our customers safe and happy, and now we are offering private cultural experiences that are customized and exciting for tourists and their families".

Following the government rules, the elite heritage tourism owner emphasized:

"We were not accepting any customer above 70 years of age. We communicated clearly that anyone with high-risk medical conditions or respiratory issues would not be entertained to join our experiential tours".

To understand how guidelines and regulations affect tourism companies, the interviewees explained the changes they enforced for their guests. For example, Platinum Heritage wanted to go beyond what was recommended to protect their guests and staff, as described by this passage below:

"We specialize in Eco-friendly safari with luxury meals. Following regulations, Safari canvas tents are submitted for electro-static sanitization, and vehicles, furniture, and fixtures are thoroughly sanitized. Each employee is trained as a COVID-19 Ambassador from Mohammed Bin Rashid University of Medicine and Health Sciences. All staff and guests will have their temperatures checked with contactless thermometers. A team and a Hygiene Manager will be responsible for checking and implementing COVID-19 protocols. The number of guests is strictly limited to one person per boat ride. Food and beverages will be served in biodegradable single-use containers, and Buffet will not be offered anymore. We have sourced sustainably produced cutlery that did not result from trees being cut down. We will add the cutlery to our bio-garden to ensure it is disposed of responsibly. Social distancing is also respected inside cars. Only open cars are used to 50% capacity. Children under 12 and adults over 60 require a private vehicle and private location as they are more at risk. Visitors over 70 years old or

with high-risk medical conditions, chronic diseases, and respiratory illnesses will not be accepted because of the current restrictions".

A participant who offered culinary tour services to tourists also cut down the number of group members to a maximum of eight. It is now offering smaller group tours that are restricted to eight people. She further explained that:

"On a tour with tourists, we do not enter any eating area without asking them to sanitize in front of us the seating area and all individual portions to be served in disposable containers. We have developed an app ensuring that visitors follow the guidelines strictly where all our cooking and payments are contactless. We also provide our guests with a free sanitation kit, including a face mask, mini hand sanitizer bottle, and a pack of wipes".

### 3.3. Organisational Adaptation and Innovation for Heritage Tourism

Further to the safety measures, as mentioned above, the Dubai Government, tourism agencies, museums, and art galleries have tried to innovate their services by developing new customized virtual technology offerings. "Museums and other attractions have gone virtual to cater to such needs and provide a safe temporary alternative for visitors to access", observes a private operator. Culture and art lovers can access movies, audio, and pictures of archaeological items, in addition to educational exercises for the younger visitors through the website and mobile App Le Louvre Abu Dhabi, with the following audio languages: English, Arabic, French, German, Hindi, Mandarin and Russian [49].

The luxury services owner reported that:

"Our customers can join a 360-degree online tour of the museum's exhibition, and The Art of Chivalry Between East and West could be appreciated through multisensory experiences".

Adapting the online innovation, the Dubai Culture and Arts Authority and Art Dubai Group joined hands to fight the impact of COVID-19 on Dubai tourism. For example, The Dubai *Ideathon* was organized on 24 April 2020, and everyone was invited to share their ideas and suggest innovations to help the cultural and art sector through tourism services [49].

The Director General of Dubai Tourism explained in detail the innovation drive:

"As we look ahead to a gradual re-opening of tourism, we will focus on the key elements that have ensured the industry's success over the past decade, creating unique value and delivering an uncompromised guest experience. To achieve this, we rely on the solidarity of our stakeholders, who have always played a pivotal role. We hope they will continue to lead from the front in positioning Dubai as a must-visit destination. Indeed, it is imperative in these unexpected times to harness our collective strengths and design effective solutions that ensure Dubai's prominence on the global tourism industry stage. Together we can redefine the future of travel".

Some participants reported being innovative in giving rather than earning, so they started free food initiatives to support the jobless and those of need during the pandemic. The local tourism expert detailed that:

"We planned to support people losing jobs and various eateries losing customers. Our team used innovation to sustain small restaurant teams by giving them orders for preparing food for those in need that were sponsored by the public. Our former customers would buy the charity gift cards, and the fees would go directly to our community restaurants to help their cash flow. Each gift card covered home-delivered meals from restaurants that tourists had previously enjoyed".

The local cultural expert mentioned the methods of adaptation and innovation:

"Tourism companies are adapting the idea to target new segments as various agencies do Zoom Seminars. I published a relevant article in the local newspaper. I explained that we have different cultures, different people. What to speak? How to speak on Zoom? Before, people did not need this, and now they need this. The new workshop on Culture Etiquette over Zoom is called 'Etiquettes of using Zoom across Cultures. Now, more people can join the program. Earlier, my trainers used to talk personally and had face-to-face sessions with a limited audience of around 25. With online sessions on Zoom, I can talk to many people in my seminars and attract them towards the Emirati Culture and heritage".

The respondent, who offers walking culinary visits in Dubai, created new virtual travel experiences for culinary seekers.

"I initiated, out of need, an online Spicy Bingo for team building, friends, and family events. It is an online and interactive game to learn about the spices sold in various spots in Dubai. I also augmented the business with an online and mobile-friendly guidebook to the historic spice souk of Dubai. Now I am also using a podcast called 'Deep Fried' supporting us to promote its services and Dubai's authentic restaurants and culinary history".

*3.4. Creative Recommendations for Heritage Tourism Post COVID-19*

The last phase of all interviews requested respondents to specify any tourism-related strategies that could be suggested to manage and protect heritage tourism from COVID-19 or any other pandemic in the future.

Insert Figure 1. Conceptual Framework: Management of Heritage Tourism During COVID-19

The Chamber of Commerce specialist suggested that:

"We must redesign public spaces, attractions, dining areas, and queuing systems to ensure social distancing. New measures could also require that all attractions implement advanced reservation systems going forward, thereby minimizing potential risks. Finally, a thorough communications plan highlighting all the above is crucial to share with all players in the industry, regardless of size".

The local business operator also expressed his ideas:

"A lot needs to be done by being very strict and assuring our tourists that they are coming to a COVID-free area. We need to create incentive programs for a safe way to transport guests from the hotels to where they want to go. Food, hotels, and attractions should be financially accessible. Tourists' trips for four days to a week are taken under control. We must prove to them that it is difficult to get sick if they follow all procedures. Moreover, if they get sick, we will take care of them. This will build our reputation as it is safe to be here".

The cultural expert thought of cultural institutions:

"Cultural institutions will need to ensure public confidence in the safety of their sites and services and, therefore, must devise a public communications strategy to attract visitors. Now we need flexibility and better timing. No matter where you are, keep your prices reasonable, even give discounts when you are available. Everyone worries about what we can or cannot do, so we address their worry and sustain our businesses".

The luxury tourism owner appreciated the current strategies of the Dubai Government and said:

"In my opinion, they (the government) are doing a great job, keeping society safe by implementing policies that will also protect tourism". I was also satisfied with actions taken to control COVID-19, "I do not have any new or different suggestions to improve the tourism practices by our government".

## 4. Discussion

A discussion of the findings and results in this paper leads toward constructing a multi-level framework by applying the four themes emerging from this study. The four themes are presented here as suggestions for the government of Dubai, the tourism industry, and tourism agencies operating at the macro, meso, and micro levels as suggested by [52]. The framework is illustrated in Table 2:

**Table 2.** Framework for multi-level health crisis recovery strategies.

| Levels of Application | Repackaging Heritage Tourism | Long-Term Safety Measures | Organizational Adaptation and Innovation | Creative Recommendations |
|---|---|---|---|---|
| **MACRO Gov and Int'l agencies** | Not applicable | Follow WHO policy and rules for businesses and tourists | Dubai's Dept of Economics and Tourism has a new section for health care and crisis management | New health care policy for residents and tourists |
| **MESO Tourism and Health industry** | Mix packaging and cooperative marketing | Follow GOVT rules, monitor tourism agencies | Tourism agencies partner with health firms and update policies | Redesign public places and new sanitization processes |
| **MICRO Tourism agencies** | Heritage tourism mixed with other popular types, vertical integration with suppliers | Apply rules to all staff members and tourists being served with heritage packages | Hire at least one health specialist to adopt and follow health regulations to protect staff and tourists | Apply social marketing to show care for the health and safety of staff and tourists |

The findings of this research indicate that public and private tourism agencies need to build new communication channels with all stakeholders of heritage tourism. All respected participants agreed with the government's plans and recovery strategies. The discussions with these experts were similar to the findings of [52] on consumer behavior related to heritage tourism. The experts explained various emergent strategies crafted and implemented in tourism services and marketing that supported the recovery of heritage tourism, which has been earlier suggested by [12]. The emergent strategies were observed to be aligned with the survival strategies illustrated in an earlier study by [20,53]. Nevertheless, they all expressed the presence of a communication gap between the government, businesses, and the people. Hence, new communication channels were required, including promotion sites, social media, and engaged people.

The diversification of tourism services based on partnerships with other service agencies was collectively mentioned by all respondents. An earlier study by [9] observed that heritage tourism could be branded as a more narrowly packaged specialized product. However, that research was conducted before COVID-19; this paper disagrees and suggests more diversified services under the brand of heritage tourism in Dubai. The tourism experts working with the government and academics applauded the government's major decision to postpone the EXPO 2020 as an emergent and recovery strategy. This strategic move by the government presents an exemplary case study on survival strategies, as explained by [20]. Furthermore, the discussion on heritage tourism sites and the overlapping of their services correlates with the selection of the three tourism-oriented heritage districts in the UAE as specified by [39].

Social media tools are commonly used and provide a quick, low-cost, and easy method of communication. The promotion sites represent the point or place of contact, such as the billboards on main roads or screens flashing around the malls, reminding people to be cautious. Similar messages could be flashed on billboards for tourists coming to the UAE at airports, hotels, malls, and famous places. These communication tactics have been crucial since the Dubai Government decided to open the doors for tourists with regulations, even while many countries were still wholly shut down as reported by [7]. This decision was

not appreciated in some international circles despite it being proved correct, even though it was contrary to the findings of other tourism researchers, including [9,19].

Recognizing the threat of pandemics related to personal hygiene and social distancing, a common thought is to change the structure, visiting procedures, washing and food facilities, and the timings for tangible heritage tourism places in Dubai. As highlighted in this study, many tourism businesses capitalized on the opportunity to create new offerings such as smaller tours, virtual socio-cultural sessions, and interactive cultural games. Even museum visits could be replaced by virtual museum tours and online interactions for museum-related kids' activities. The COVID-19 crisis could catalyze the acceptance of a new travel experience. Tourism policymakers could help with financial resources for innovation and virtual infrastructure for heritage tourism organizations and entrepreneurs to adopt Smart Tourism technologies to manage any future health crisis. Finally, public–private partnerships must play a key role in entrepreneurial start-up funding to support smart technologies and innovative ways of managing heritage tourism. The multisensory experience based on online tourism offerings is supported by the suggestions presented by [50]. However, relying on online visits differs from the findings of [30] that entrenched the emotional value of heritage tourism with physical visits rather than virtual experiences.

## 5. Conclusions

As mentioned previously in this study, the purpose of this research is revisited, and is aligned with the overall findings:

(1) What *business strategies* of heritage tourism at the *macro, meso, and micro levels* were changed in tourism organizations due to COVID-19?

This objective was successfully attained as the findings illustrated critical issues at all three levels for heritage tourism during COVID-19. The macro-level influences illustrated how such decisions shaped the meso and micro outcomes.

(2) What changes were implemented for heritage tourism management strategies during and after the COVID-19 pandemic?

As recognized during the data analysis processes, functional changes were implemented incrementally, as identified during the COVID-19 pandemic restrictions. This was implemented in Dubai airports opened for particular countries but required all passengers to undergo PCR testing before arrival. This progressed to further restrictions, which required approved vaccinations before their travel to the UAE.

(3) How have tourism organizations managed the effect of new initiatives in response to the pandemic?

The challenges were managed to mitigate the effects through the four strategies that were identified, as shown in Figure 2. These strategies comprised: (i) Repackaging heritage tourism, (ii) Long-term safety measures for tourists, (iii) Organizational adaptation and innovation, and (iv) Creative recommendations.

(4) What tourism policy measures could be recommended to the policymakers for post-COVID-19 business recovery?

Based on the policy measure that is to be proposed at the macro level, tourism policymakers could aid financial resources for innovation and virtual infrastructure for heritage tourism organizations. At the meso level, public–private partnerships will continue to play a vital role in entrepreneurial start-up funding to support smart technologies and innovative ways for effectively managing heritage tourism. As there is an increasing demand for entrepreneurs to adopt technologies, there is a need at the micro level for developing digital literacy and utilizing such innovations, i.e., Smart Tourism technologies, which have been proposed to manage businesses and operations during unprecedented and unpredictable events, such as any future pandemics and beyond.

This qualitative paper presents a theoretical model based on four interrelated themes: (1) Repackaging heritage tourism, (2) Long-term safety measures, (3) Organizational adap-

tation and Innovation, and (4) Creative recommendations. These themes are analytically mapped to the three tiers of analysis (macro, meso, and micro) of management levels, suggesting strategies to improve the management of heritage tourism in Dubai at each level. Hence, the study evaluates the impact of COVID-19 on heritage tourism in Dubai and presents strategic solutions for health management strategy, practice, and policy in Dubai.

Some of the critical limitations of this study are attributed to the hesitation among participating firms to share their successful ideas, which were considered sensitive due to restrictive organizational policies. Furthermore, a few participants were interviewed as a qualitative study, so we hesitated to generalize these findings. As this study presents only the supply side of the investigated phenomena, a more comprehensive understanding of this issue could have been explored. Considering the qualitative design of the study and the phenomena investigated during post-pandemic period, it is potentially subject to respondent subjective recall bias.

This research creates research opportunities to study the impact of COVID-19 on heritage tourism in various destinations around the globe. Interviews in this study highlighted the shift in heritage tourism service offerings that helped businesses operate and respect the new health and safety measures. This research could be replicated in other countries, and the findings could be compared to reach more generalized solutions to pandemic management in the future. The novelty of this study is inherent in its contextualization of an under-explored area, concurrent analysis of a dynamic and lucrative sector, and methodological advancement through the embellishment of techniques. Constructed upon the consequent findings of this research, a contextualized framework is proposed that complements tourism theory and delivers credible implications for researchers, government planners, and tourism providers.

The four themes illustrated in Figure 1 could be used to understand observations of local and foreign heritage tourists regarding the impact of COVID-19 on UAE heritage tourism management strategies. Based on the implications of these findings, future research could take into consideration the following: (1) studies could target a different sample of government officers involved with COVID-19 recovery strategies in a qualitative study; (2) a quantitative study with tourists could be a natural extension of the research, where a positivist paradigm could indicate more quantified findings for future tourism planning; and (3) as this research focuses on the supply side, further studies could explore the demand side and the impact of this phenomenon.

**Author Contributions:** Conceptualization, N.Y. and F.H.; methodology, G.N.; software, N.Y. and F.H.; validation, N.Y., F.H. and G.N.; formal analysis, N.Y.; investigation, F.H.; resources, F.H.; data curation, N.Y.; writing—original draft preparation, N.Y.; writing—review and editing, N.Y.; visualization, G.N.; supervision, N.Y. and F.H.; project administration, N.Y.; funding acquisition—N.Y. All authors have read and agreed to the published version of the manuscript.

**Funding:** FCAS, Canadian University Dubai.

**Data Availability Statement:** Not applicable.

**Conflicts of Interest:** The authors declare no conflict of interest.

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
