# Peer review of "Envisioning the Future of Heritage Tourism in the Creative Industries in Dubai: An Exploratory Study of Post COVID-19 Strategies for Sustainable Recovery"

_heritage, doi:10.3390/heritage6060242_

Round 1
Reviewer 1 Report
The contents:
164 - you write about predictions concerning 2020 but it has passed already so the data should be updated;
the second part of the 2.3. supchapter is actually about the methods so it should not have this title, but be separated; on the other hand - the text is a bit illogical with few repetitions like in 186 and 196 lines, while the exact data collection details are ot provided: number of people interviewed and the structure of the group (share of representatives of different sectors) and the period of research;
There are some minor flaws like:
Abstract: first word in bold;
lines 18-20 - after numbers either you start with large or with a small letter;
as far as I see the in-text citation should follow the order of appearing in the text not otherwise;
63 - you mean SARS?
line 95 and more - I do believe you should not start the sentence with [11] in-text referencing but with the name of the authors
116 - I do not agree with the statement that get together locations are heritage attractions.
127 it does not make sense to speak here of general types of heritage sites after already mentiooning two such UNESCO heritage sites in Dubai
137 - reincarnation?
269 - ends with full stop not ":"
411 - religiously?
Author Response
Dear Reviewer,
Thank you for providing your comments on our paper. Your comments have helped us effectively improve the quality of our paper.
Thank you
Best wishes,

Reviewer 2 Report
Dear authors, I have reviewed the document "Envisioning the Future of Heritage Tourism in the Creative Industries in Dubai: An Exploratory Study of Post Covid-19 Strategies for Sustainable Recovery", which was intended: (i) Inconsistencies for Heritage Tourism, and (ii) Appreciation for Heritage Tourism. As a result, subsequently, four multi-tiered themes (macro, meso, and micro level) emerged as a response to the current challenges which are (i) Repackaging heritage tourism, (ii) Long-term safety measures for tourists, (iii) organizational adaptation and innovation, (iv) creative recommendations.
The following are the main observations:
1. The authors should not repeat the words of the title in the keywords of the manuscript.
2. The abstract of the manuscript should be modified, the objective of the research should be very clear and precise, in the current state it is not clearly understood.
3. In the first paragraph of the manuscript [43][45] is cited, I do not understand why this appears here.
4. The first paragraph of the introduction should focus on describing the current situation of tourism at all levels.
5. The first paragraph of materials and methods is out of context in this section, this should be in the introduction. A quote [51] appears here, this is not correct.
6. The authors should review the order of citations, the current state is deficient, impossible to think of publishing in this way.
7. The methodology section does not present a design according to a research, it looks more like introduction paragraphs with some questions...
8. It is said that they are interviews to experts, perhaps I am trying to carry out a technique of expert judgment or semi-structured interviews, the methodology could improve notably if the authors order the ideas. Use tables with the experts, diagram of the methodological design.
9. In the discussion section should present citations, the essence of a good discussion should focus on comparing your results with other studies and discuss these similarities or differences.
10. I suggest that in both the discussion and conclusions the authors should generate a paragraph for each of the objectives established in the document.
11. Finally, it is necessary to clearly and punctually state the limitations of the research and where future research should be directed.
I recommend the authors to review the grammar and especially the structure and style of writing.
Author Response

(The authors gave the same response as above.)

Reviewer 3 Report
Dear Author (s)!
The manuscript is very interesting. Touches on a current topic.
Small recommendations:
- The purpose of the research should be clearly defined.
«the purpose of this research is to investigate suggestions from experts on heritage tourism strategy and policy in Dubai, after the extremity of COVID-19». «The focus of this paper is to study the strategies to manage the COVID-19 pandemic for heritage tourism in Dubai». «This qualitative research explores the contextual challenges associated with the pandemic and presents a range of strategies».
- It is appropriate to finalize the abstract.
The abstract must include sufficient information for readers to judge the nature and significance of the topic. The abstract should contain the main idea of the paper, the subject and the goal of the research, methods used, hypotheses, research results and a brief conclusion.
- It is worth adding a «literature review» section. The literature review should also be concluded with 2-3 summarizing sentences. Then the purpose of the research should be formulated. After that, you should state (if you foresee it) hypotheses.
- A certain part of the information from the «Materials and Methods» section should be in the literature review.
-There is no logical transition from 2.1. Dubai and Heritage Tourism to 2.2. Managing Past health crisis.
- To reformat and systematize the material of Section «2. Materials and Methods».
- Pay attention to the non-scientific style of presenting the material in the "Results" section:
«The specialist ...elaborated:», «...private operator said:», «The Director... explained», «Similarly, an executive in Emirates highlighted», «Another respondent.... explained», «Similarly, the founder of an experiential company stated», «The owner of an elite and luxury experiential travel agency expressed».
- It would be appropriate to present information about the considered strategies in graphic form.
- How to understand "heritage tourism COVID-19"?
- Maybe the author's opinion is not completed? «Insert Figure 1. Conceptual Framework: Management of Heritage Tourism During 472 COVID-19».
- In addition to presenting theses with the opinions of experts, the author(s) of the article should analyze and summarize the given material. Critical analysis is absent. After presenting the graphic material in Fig. 1, it is necessary to analyze the presented data.
- The conclusions should be finalized, significantly expand. Conclusions must be consistent with the evidence and arguments presented and correspond to the main question posed. The strategies for sustainable recovery should be clearly defined and presented in the conclusions.
- The statement «This qualitative paper contributes to the theory of tourism and heritage by developing a theoretical model based» is strange. The quality of the article is evaluated by the reader, not the author (s). It is not appropriate to use the constructions «the theory» - «theoretical model», «... strategic solutions for the health management strategy».
I am grateful to the author (s) for the interesting material they have prepared.
Best regards
Author Response

(The authors gave the same response as above.)

Round 2
Reviewer 2 Report
Dear Author, I have reviewed the paper "Envisioning the Future of Heritage Tourism in the Creative Industries in Dubai: An Exploratory Study of Post Covid-19 Strategies for Sustainable Recovery", which aimed to explore the contextual challenges associated with the pandemic and presents a variety of strategies. The author prior to further processing his paper should focus efforts on the following comments:
1. Authors in the abstract should maintain a basic, orderly order. The abstract in the current state mixes methodology, objective, then methodology again. Authors should generate two sentences of introduction (What if any), one problem sentence, one sentence of objective and/or research questions, one sentence on the design, approach and tools used in a very timely manner. One or two sentences on the results achieved and conclusions-challenges.
2. In the introduction devoting two paragraphs to COVID-19 seems excessive to me. A second paragraph should be devoted to the tourism industry in times of covid, a third paragraph on the tourism segment under study.
3. The paragraph on line 87 should be the last paragraph of the introduction.
4. Eliminate the literature review section, just keep the introduction with 6 or 7 paragraphs at most.
5. Why do you repeat the objective in the first paragraph of the methodology, I don't understand this. In a first paragraph you should address the research design, approach, tools and techniques used in the form of an introduction.
6. In the previous review I stated that the methodology section looks like an introduction.
7. The methodology should be redesigned is very extensive and is not understood in narrative form. Synthesize the ideas.
It is impossible for me to review the requested changes because the authors are brief in their answers and do not explain the changes made. Additionally, if changes were made, they are not evident or I do not believe that the changes are only those highlighted in yellow.
Finally, I recommend that the authors redesign the research, clearly generate two or three objectives and/or research questions, with this they will be able to properly elaborate the methodology, results, discussion and conclusions.
Check this structure https://www.mdpi.com/2071-1050/13/8/4432 take it as an example. I don't want you to cite it, just consider the design of the methodology, look at the order between the last paragraph of the introduction, methodology and results.
--
Author Response
Dear Reviewer
Thank you for your comments. Please kindly find attached response to reviewer 2 table.
Your comments have been precious in improving the quality of our work.

Reviewer 3 Report
Dear Authors!
Thank you for considering the suggestions and improving the quality of the article.
All the best,
Author Response
Thank you.